# How Confinement and Back to Normal Affected the Well-Being and Thus Sleep, Headaches and Temporomandibular Disorders

**DOI:** 10.3390/ijerph20032340

**Published:** 2023-01-28

**Authors:** Juan Ignacio Rosales Leal, Cristian Sánchez Vaca, Aleksandra Ryaboshapka, Félix de Carlos Villafranca, Miguel Ángel Rubio Escudero

**Affiliations:** 1Department of Stomatology, Prosthodontics & Orofacial Pain Section, School of Dentistry, University of Granada, 18071 Granada, Spain; 2Department of Surgery and Medical-Surgical Specialties, Orthodontics Section, Faculty of Medicine, University of Oviedo, 33003 Oviedo, Spain; 3Department of Computational Science and Artificial Intelligence, School of Computer and Telecommunications Engineering, University of Granada, 18071 Granada, Spain

**Keywords:** COVID-19, confinement, new normal, questionnaire, well-being, sleep, headaches, temporomandibular disorders

## Abstract

The COVID-19 pandemic is having negative consequences not only for people’s general health but also for the masticatory system. This article aimed to assess confinement and its new normal impact on well-being, sleep, headaches, and temporomandibular disorders (TMD). An anonymous survey was distributed to a Spanish university community. Participants completed a well-being index (WHO-5), a questionnaire related to sleep quality (the BEARS test), a headache diagnostic test (the tension type headache (TTH) and migraine diagnosis test), and the DC-TMD questionnaire. Questions were addressed in three scenarios: before confinement, during confinement, and the new normal. A total of 436 responses were collected (70% women, 30% men). A reduction in well-being and sleep quality was recorded. Respondents reported more TTH and migraines during and after confinement. Overall, confinement and return to normal did not increase TMD symptoms, and only minor effects were observed, such as more intense joint pain and a higher incidence of muscle pain in women during confinement. Reduced well-being is correlated with sleep quality loss, headaches, and TMD symptoms. This study provides evidence that pandemics and confinement might have had a negative impact on population health. Well-being was strongly affected, as were sleep quality, depression risk, TTH, and migraine frequency. In contrast, the temporomandibular joint and muscles showed more resilience and were only slightly affected.

## 1. Introduction

COVID-19 is posing a great challenge to humanity. The world is changing rapidly to adapt to the new situation. There are important global consequences for the economy and health management that affect people directly. The Corona outbreak created a world of ambiguity, loss of control, and uncertainty [1]. Due to the initial lack of information on the SARS-CoV-2 virus, ignorance of its routes of spread, lack of effective treatment, initial lack of vaccine, and rapid saturation of world hospitals, most countries chose to establish a partial or total confinement of the population in order to try to stop the spread of the disease and the collapse of the health system [2,3]. After confinement, strong rules were established to stop the pandemic over a long period of time [4]. From this moment on, the daily routine of the population was completely altered, as it had never been seen before, and its consequences are still present.

One of the most noticeable effects of the pandemic has been mental and emotional. The pandemic has led to a significant increase in anxiety, which in many cases has led to depressive situations [5,6,7,8,9]. Several studies determined that social isolation, suspension of work and educational activity, the economic crisis derived from the pandemic, and fear of contagion affected the physical and mental health of the population [5,6,10]. An increase in stress, anxiety, and depression, sleep impairment, and even post-traumatic stress have been detected in the population during and after the period of confinement [5,6,10]. As a consequence, the pandemic is having a very detrimental effect on the well-being of the population [11]. The pandemic also had negative effects on sleep [8,9]. The prevalence of sleep problems during the COVID-19 pandemic has increased from approximately 15% of people before the pandemic to approximately 40% [12]. Sleep is greatly affected when a person’s environment is not favorable. Sleep is essential for emotional, physical, and cognitive well-being [13]. Sleep loss is closely associated with cognitive impairment and mood changes and has negative impacts on quality of life, mood, cognitive function, and health [14]. Its consequences are still visible, and its extent and relationship with other parameters need further study.

Additional stress during the pandemic may have increased the incidence of primary headaches. Primary headaches are disorders that exist with no apparent underlying cause and with recurrent or persistent head pain [15], and the most frequent are tension-type headaches (TTH) and migraines [16]. In general, primary headaches are accompanied by a loss of well-being in the sufferer [17]. Psychological factors play a crucial role in many patients and are behind the genesis of a TTH, and stress or sleep disturbances can trigger the cephalalgia [18,19]. In the case of migraines, there is a significant comorbidity with depression and anxiety [20] or with poor sleep quality [21].

TMD is a group of painful and/or dysfunctional conditions of the masticatory muscles, temporomandibular joint (TMJ), and associated structures [22,23]. The overall prevalence of temporomandibular joint disorders is approximately 31% [24]. TMD have a multifactorial etiology, including as potential risk factors: prolonged use of mastication muscles, grinding and clenching, some malocclusion, repetitive trauma at the TMJ, psychological disorders, cervical posture, or even the position of the cervical spine [22,23,25].

Psychological factors such as stress, depression, or anxiety are part of the etiology of TMD [22]. Emodi-Perlman et al. [26] have found an association between the psychological effects of the pandemic and the increased intensity of TMD and bruxism symptoms among the general populations of Poland and Israel [26]. A study carried out in Italy has determined that the pandemic and confinement have been able to trigger the appearance of TMD and the worsening of symptoms in already diagnosed patients [27]. Another study carried out in dental students in Turkey has associated the increased prevalence of temporomandibular disorders with increased stress, anxiety, depression, and altered sleep levels [28]. But more studies are needed to determine the effect of the pandemic on temporomandibular disorders and its possible association with other factors.

The biopsychosocial model describes the complex interactions between biology, psychological states, and social conditions that cause and/or maintain function or dysfunction [29]. This model describes the complex interactions between biology, psychological states, and social conditions that can cause disease or maintain health. This model is included in the TMD diagnosis. There is a dual-axis system for the diagnosis of TMD: Axis I (clinical examination) and Axis II (pain-related disability); axis II evaluates relevant behavioral, psychological, and psychosocial functioning [30].

Comorbidities are of great interest in the field of orofacial pain. Headaches, TMD, and sleep are strongly associated. Poor sleep quality increases the risk of TMD, and people who develop TMD have impaired sleep quality [31,32]. Sleep is also related to headaches, and headaches are related to TMD [33,34]. Comorbidities among these health problems might be analyzed. The comorbidity of these health problems needs to be analyzed, and more studies are needed to better understand this association in times of pandemic.

Many of the health consequences of the pandemic can be explained by the biopsychosocial model. Confinement and the new normal are strong social conditions that have affected the psychological state and thus may have generalized biological problems. Further studies are needed to clarify the effects of confinement and the pandemic on people’s well-being and their effect on sleep, headaches, and TMD. The comorbidity between these four variables should also be analyzed in the COVID-19 context. This study hypothesizes that social factors (confinement and pandemic) have affected psychological states (well-being) with biological repercussions (sleep, headaches, and TMD). The objective of this study was to assess the lockdown and new normal impact on well-being, sleep, headaches, and TMD among a Spanish university population.

## 2. Materials and Methods

### 2.1. Study Design

This was a cross-sectional study. An indiscriminate anonymous online survey was prepared using the “Google Forms” tool. The questionnaire included a general section with demographic questions and specific sections with questions about TMD, headaches, sleep, and well-being. The same questions were done under three different time scenarios: pre-lockdown (before the pandemic); lockdown (time spent during confinement); and post-lockdown (also called “the new normal” or “back to normal”). Before answering the questionnaire, participants were encouraged to recall these three different scenarios. The study protocol was approved by the Ethics Committee of Granada University (Spain) (registration number 2250/CEIH/2020). The questionnaire was distributed via email and social networks to members of the University of Granada. Data collection was carried out in September–October 2020. A questionnaire was sent once to the participants.

### 2.2. Participants

Members of Granada University (Spain) were invited to participate in this study. The inclusion criteria were to be a university student or university worker and not have had COVID-19. All participants signed an “informed consent” in which they were provided information on the development and objectives of the survey, in addition to allowing the data to be stored anonymously. The target population was 80,000 (60,000 students and 20,000 workers). A minimum sample size of 383 participants was calculated (confident level: 95%; confident interval: 5%).

### 2.3. Content

The survey was structured into five sections:

#### 2.3.1. Section 1. Demographic Data

This section included age, gender, occupation, illnesses, drug intake, student or worker status, and destination. Respondents who were or had been ill with COVID-19 were eliminated from the study.

#### 2.3.2. Section 2. Well-Being Index

A validated Spanish version of the World Health Organization’s Five Well-Being Index (WHO-5) was used [35]. WHO-5 is a self-administered five-item scale. Each item assesses the degree of positive well-being during a period of time on a six-point Likert scale graded from 0 (at no time) to 5 (all of the time); the total score ranges from 0 to 25, with high scores indicating an increased sense of well-being. Respondents who scored less than 13 points were specifically recorded.

#### 2.3.3. Section 3. Sleep

Sleep was assessed with the BEARS sleep screening tool [36,37]. BEARS includes five questions on sleep problems: excessive daytime sleepiness, awakenings during the night and problems returning to sleep, insufficient regularity and duration of sleep, and snoring with a positive or negative response. To quantify the results, responses were scored as 0 if negative and 1 if positive, and a total of 5 points could be scored. The higher the score, the worse the quality of sleep. In addition, specific questions about headaches on awakening or bruxism during sleep (possible sleep bruxism) and wakefulness (awake bruxism) were included: “Do you have headaches when you wake up in the morning?”; “Are you aware of clenching or grinding your teeth when you sleep?”; “Are you aware of clenching or grinding your teeth while awake?”

#### 2.3.4. Section 4. Headaches

This section was elaborated based on the International Headache Society (IHS) diagnostic criteria for TTH and migraine [16] and included questions about the presence or not, the frequency, the intensity, the quality, the location, the related symptoms, the interference with routine activities, and the aggravation with physical efforts. TTH and migraines were categorized as frequent episodic (FETTH) or infrequent episodic (IETTH). Unclassifiable headaches based on these questions were categorized as non-diagnosable.

#### 2.3.5. Section 5. Temporomandibular Disorders (TMD)

The “symptom questionnaire” of the diagnostic criteria for temporomandibular disorders, which corresponds to the sections “joint pain and intensity,” “muscle pain and intensity,” “mandibular joint noises,” and “jaw lock” (DC/TMD) [30], was applied. The questionnaire included both myogenous and arthrogenous TMD, according to DC/TMD [30]. The limitation scale of mandibular function (JFLS-8) [38] was also included.

### 2.4. Statistical Analysis

In this study, different statistical analyses have been carried out depending on the characteristics of the responses provided by the respondents. In the case of numerical responses—for example, the intensity of jaw pain—the Student’s *t*-test of paired measurements was used. Paired-sample *t*-tests have higher statistical power than independent-sample Student’s *t*-tests. In those cases where the response was categorical with only two categories—for example, the presence or absence of a headache—contingency tables were created for the responses before confinement, during confinement, and after confinement. The tables obtained were analyzed using Fisher’s exact test. In the case of the temporomandibular joint, the McNemar test was used. When the responses were categorical but ordinal—for example, the question about the well-being of the respondents—the responses were assigned to a Likert scale, and subsequently the Student’s *t*-test for paired samples was used. There has been some discussion in the literature about the convenience of using this test when working with Likert scales, but it is considered that under normal circumstances it provides robust results [39]. In the case in which the question gave rise to non-ordinal categorical responses—for example, the type of headache—Pearson’s χ^2^ test was used to compare the distributions obtained before confinement, during confinement, and after confinement. Associations between pairs (TMD-joint pain intensity [TMD-J], TMD-muscle pain intensity [TMD-M], headache (pain intensity), sleep (Bears score), and well-being) were obtained with Spearman’s rank correlation coefficient. A Spearman rho between variables and a 95% confidence interval were calculated. All statistical analyses in this study were performed with the R statistical software package [40] using version 3.5.3 (The R Project for Statistical Computing, Vienna, Austria).

## 3. Results

The survey was answered by 436 people, of whom 305 were women (70%), and 131 were men (30%). The age range was between 19 and 70, with the majority being young people. The mean age was 31.6 years, with a standard deviation of 14. Of the total number of participants, 298 were students; the remaining 138 people belong to the rest of the university community (teachers, workers, etc.).

The WHO-5 well-being index results are shown in Table 1. The pandemic negatively influenced the well-being of the population. During confinement, the WHO-5 index decreased. At new normality, well-being was improved over the period of social isolation, but previous state was not recovered. People who scored less than 13 points followed the same tendency. There were more people with less than 13 points on lockdown, and this percentage was lower in the new normal but higher than at the beginning. There were gender differences in the group of people with scores less than 13. At the beginning, there were no gender differences. However, confinement led to more women scoring below 13. In the new normal, this percentage was reduced, but more women than men scored below 13.

Sleep data are shown in Table 2. People reported more bedtime problems, awakenings during the night, and trouble getting back to sleep. Excessive daytime sleepiness was also increased. Regularity and duration were also altered and reduced. On the contrary, snoring was not altered. BEARS scoring showed a sleep quality reduction during pandemic. Lockdown increased scoring and returned it to normal, improving sleep quality but not at the same level as before the pandemic.

Table 3 shows the data on waking headaches and bruxism. Confinement increased waking headaches, and values were maintained at normal. The prevalence of possible sleep bruxism and awake bruxism did not change in men. In contrast, women increased their possible sleep and awake bruxism in the pandemic. Women’s sleep bruxism increased during lockdown, and its incidence was maintained after. Women’s awake bruxism was increased only in the new normal.

Results for headaches are shown in Table 4. The pandemic did not alter the prevalence of IETTH but did alter the prevalence of FETTH. In females, more FETTH were observed only during confinement. In men, FETTH increased with confinement, and values remained higher at the new normal. Regarding migraines, they only increased in women after conception and afterwards.

The TMD results are shown in Table 5. Pandemic or lockdown did not increase the TMJ pain prevalence. Only women who initially had joint pain experienced an increase in joint pain intensity during confinement. After lockdown, joint pain intensity was the same as at the beginning. TMJ pain was more prevalent in women than in men. More women had muscle pain during lockdown, but the intensity was always the same. Confinement did not affect men’s muscular pain, and men had lower muscle pain than women. Both men and women felt more joint noises during confinement. Noises were more frequent in women than in men. No changes were found in jaw blocking or mandibular function.

Table 6 shows the association between well-being, sleep quality, headache intensity, and TMD (joint and muscle pain). There is a significant association between all the parameters analyzed.

## 4. Discussion

Overall, it was detected a significant effect of the pandemic on the well-being sensation of the subjects. The WHO-5 used in this study is a short, 5-item index designed to assess the level of emotional well-being [41]. The WHO-5 has been widely used to screen for depression in primary care [42]. This test is negatively correlated with depression [43,44]. In addition, there is a positive correlation between the WHO-5 index values and the quality of life [8]. In this survey, respondents perceived a significant reduction in well-being during lockdown and after. Then, the pandemic reduced the quality of life and increased depressive risk.

It is of particular interest to analyze WHO-5 scores below 13. A cut-off point less than 13 indicates poor well-being and has been associated with depression [26]. Before the lockdown, subjects scored below 13 only represents 2%. However, during and after lockdown, it was greatly increased to a fifth of the subjects. This effect was much more intense in women than in men. With this data, we can affirm that the pandemic and confinement affected people’s mental health and increased their depression risk. Published studies obtained similar results: stress, anxiety, and depression increase among the population during and after confinement [6,9,11,45,46].

Two COVID-19 impact indicators, specifically daily SARS-CoV-2 infection rates and reductions in human mobility, were associated with an increased prevalence of major depressive disorder [10]. These two COVID-19 impact indicators incorporate the combined effects of the spread of the virus, lockdowns, stay-at-home orders, decreased public transport, school and business closures, and decreased social interactions, among other factors [10]. Females were affected more by the pandemic than males [10,45], and younger age groups were more affected than older age groups [10,47]. Besides, it has been widely demonstrated that situations of anxiety or depression favor the appearance of physical diseases [8,48]. Perhaps, apart from the viral disease itself, one of the most serious effects of the pandemic has been on mental health.

The pandemic produced a modification of sleep habits and a deterioration of sleep quality. In this study, a greater difficulty in falling asleep as well as an increase in night-time awakening and difficulty in going back to sleep were determined. Daytime tiredness also increased during the day. Other studies found a decrease in the quality of sleep and insomnia in the population, as well as a change in sleep schedules during and after confinement, especially in students and young people [47,49]. In this regard, an association between circadian rhythm abnormalities caused by home quarantine during the COVID-19 outbreak and mental problems among university students was also pointed out [50]. Clearly, the new situation and the uncertainty caused by the pandemic lead people to experience more stress or anxiety, which directly affects sleep.

Alternatively, an important habit change was detected during lockdown and after. The timing of sleep or wake-up was 1–2 h later than in school, and 78.7% of the students spent 2 h/d or more on the computer or watching TV/video programs during confinement [50]. This could justify the daytime tiredness or the problems falling asleep at bedtime.

In addition, sleep alteration has important repercussions on mental health and can lead to a more depressive state. It was determined that sleepiness and sleep debt mediated the relationship between short sleep or evening-type preferences and depression and anxiety risk in university students [51]. In our study, the WHO-5 index indicated an increase in depression risk in 20% of the population, which is concordant with this.

Headaches were strongly affected by the pandemic. Women and men with frequent episodic TTH increased the frequency during lockdown and after. Men and women also suffered more migraines during confinement. However, in the new normal, men returned to baseline situations, but women continued to suffer more migraines. In the case of migraines, gender has a strong influence, with women suffering more migraines than men. Among the factors that can trigger both TTH and migraines are stress, anxiety, depression, or sleep disturbances [18,19,20,21,33]. Emotional situations have been and continue to be experienced that can act as triggers for headaches. Loss of well-being or sleep disturbances may also contribute to explaining the increased prevalence of headaches. Other studies have also found an increase in headaches and migraines as a consequence of the negative impact of the pandemic [52,53]. On the contrary, people with infrequent episodic TTH (once a month) were not affected. Such people are very unlikely to suffer from headaches. Thus, the situation experienced and the possible stress suffered or emotional triggers are not enough to produce more headaches.

Interestingly, confinement and the new normal increased the number of subjects that woke up with headaches. Morning headache prevalence in the general population is nearly 8% and can be related to many other causes [54]. For example: sleep disorders, bruxism, or sleep apnea [55]. Headaches present upon awakening were strongly related to mood disorders [56]. Even a chronic morning headache can be a good indicator of major depressive and insomnia disorders and is not specific to sleep-related breathing disorders [57]. Frequent morning headaches are associated with disturbed sleep and daytime sleepiness [58]. Subjects of this study had sleep problems and a reduced well-being index as a consequence of the pandemic. These two facts could be related to morning headaches.

One of the aims of this study was to determine the influence of confinement and return to normal in temporomandibular disorders. The effect was low and only evident in women. Women with a previous painful joint experienced more pain intensity during confinement or the new normal. Also, there were more women with muscle pain, although it was of the same intensity. Other studies found similar results, but in all the population; people previously diagnosed with temporomandibular disorders, as well as those who reported greater stress and anxiety during the pandemic, perceived their symptoms as more severe compared to the previous situation [26]. The masticatory system and temporomandibular joint have a lot of adaptive potential and a great capacity for regeneration and healing [59]. This could explain its low deterioration in a situation as particular as the pandemic. However, under pressure, an unhealthy joint could greatly increase the symptoms [59]. The role of gender was expected because most TMD patients worldwide are women [22,23]. In these cases, rehabilitative approaches might be effective in reducing pain in muscle-related TMD patients [60]. Compared to arthrogenous TMD, which appears to be a localized phenomenon, myogenous TMD may present overlapping features with other disorders, such as fibromyalgia and primary headaches, characterized by chronic primary pain related to dysfunction of the central nervous system, probably through the phenomenon of central sensitization [34,61].

Confined people noticed more TMJ noises than before and after lockdown; as always, they were more frequent in women. The clicking prevalence was similar to other studies [62,63]. The reason why respondents observed more noise when confined is not clear. Perhaps, during confinement, people had more time to think and were more aware of joint noises than before and after lockdown. Moreover, mood and personality may influence the perception of clicking [63]. In fact, it is widely accepted that stressful life events can lead to TMD [17]. The stress suffered during confinement may also lead to some overloading of the joint and increased joint noise. Or, quite simply, an anxious situation may have made the clicking sounds more noticeable.

Sleep or awake bruxism is a factor that could be responsible for TMD symptoms. There are some researchers who found a bruxism increase during the pandemic and so TMD aggravation [22]. It is important to note that questionnaires can only indicate a possible case of bruxism because only the patient’s response is registered [64]. In our study, only women’s sleep and awake bruxism were increased during and after confinement. In contrast with sleep bruxism, awake bruxism is more conditioned by anxiety or depressive states, and it is most often accompanied by muscle pain [65]. This may contribute to the increased prevalence of TMD symptoms in women. Additionally, it also could be related to waking up with headaches incessantly.

One of the limitations of this study was recall bias. Self-reported data collected through surveys is a key element in a wide range of research and can provide valuable information on social, economic, or health issues [66]. The problem is the accuracy of recall, which is conditioned by the characteristics of the exposure of interest, such as the degree of detail, the importance to the respondent, and the time period involved [67]. Interview technique and study protocol, including questionnaire design and respondent motivation, also play an important role and are under the control of the researcher [67]. One way to control for confounding factors is to assign respondents versions of the same question that differ only in the attachment period over which they are asked to recall past use [68]. In terms of temporal recall, non-significant differences have been obtained in the mean values of monthly measurements of low back pain compared to retrospective quarterly and yearly measurements at group level [69]. One year of recall is adequate to assess a health problem; although the total amount of recall error is larger for a one-year period, errors are more evenly distributed between over- and under-reporting [68]. The pandemic has been a very intense and easy experience for people to remember. People were motivated to answer the questionnaire, and their recall time was between 6 and 12 months. In addition, the questionnaire was designed with general questions that were not very detailed, easy to remember, and the same for each period.

Another limitation of the study was that the survey was done online, and a clinical examination was not carried out to detect and corroborate the results obtained. Furthermore, the sample has been restricted to members of the university community (university students and workers), so it cannot be fully extrapolated to the general population. Another important limitation of this study is that the analyzed associations were not adjusted for confounding variables. Future studies should be done with a larger sample of the entire population and include a clinical examination. In addition, future studies should include telemedicine to more easily monitor aspects such as well-being, anxiety, and sleep.

However, in general, this study contributes to demonstrating how a disruptive social situation can unbalance the mental and physical health of the population. Moreover, it has been demonstrated the interconnections between all the parameters evaluated. Loss of well-being affected sleep and headaches; decreased sleep quality increased headaches and reduced well-being; and TMJ or masticatory muscles were affected when well-being or sleep quality were reduced as well as when headaches were increased.

## 5. Conclusions

Despite the limitations of this study, the results could reveal information relevant to understanding the effect of a global health problem and a new stressful social situation in the orofacial field. This study provides evidence that pandemics and confinement might have had a negative impact on population health. Well-being was strongly affected, as were sleep quality, depression risk, TTH, and migraine frequency. In contrast, TMJ and muscles showed more resilience and were only slightly affected.

## Figures and Tables

**Table 1 ijerph-20-02340-t001:** WHO-5 well-being index.

		Pre-Lockdown	Lockdown	Post-Lockdown	*p*-Value		
					Pre vs. Lockdown	Lockdown vs. Post	Pre vs. Post
Total ^1^ (score 0–25) (mean[SD])	Total	18.84(3.14)	15.22(3.75)	16.47(3.54)	***	***	***
	Women	18.73(3.11)	14.92(3.78)	16.18(3.51)	***	***	***
	Men	19.11(3.21)	15.91(3.58)	17.16(3.53)	***	***	***
Score less than 13 ^2^ (%)	Total	4.4	21.3	12.8	***	***	***
	Women	4.3	25.2	14.8	***	**	***
	Men	4.6	12.2	8.4	*	ns	ns

^1^ T-Student test; ^2^ Fisher Test. SD: Standard deviation. *** *p* < 0.001; ** *p* < 0.01; * *p* < 0.05; ns: non-significant (*p* > 0.05). *n* = 436.

**Table 2 ijerph-20-02340-t002:** Sleep BEARS test.

Parameters (Yes %)		Pre-Lockdown	Lockdown	Post-Lockdown	*p*-Value		
					Pre vs. Lockdown	Lockdown vs. Post	Pre vs. Post
B: Bedtime problems ^1^	Total	27.1	59.6	45.4	***	***	***
	Women	28.2	65.2	50.8	***	***	***
	Men	24.4	46.6	32.8	***	**	ns
E: Excessive daytime sleepiness ^1^	Total	36.7	54.6	54.8	***	ns	***
	Women	40.0	59.0	60.0	***	ns	***
	Men	29.0	44.3	42.7	**	ns	**
A: Awakening during the night and trouble getting back to sleep ^1^	Total	12.2	36.0	29.4	***	**	***
	Women	12.8	39.0	33.4	***	*	***
	Men	10.7	29.0	19.8	***	*	*
R: Regularity and duration of sleep ^1^	Total	51.6	61.5	63.5	***	ns	***
	Women	54.4	65.9	66.6	***	ns	***
	Men	45.0	51.1	56.5	ns	ns	*
S: Snore ^1^	Total	30.5	31.0	31.2	ns	ns	ns
	Women	23.3	23.9	24.3	ns	ns	ns
	Men	47.3	47.3	47.3	ns	ns	ns
Score B + E + A + R + S (1–5) ^2^ (mean[SD])	Total	1.58(1.37)	2.43(1.61)	2.24(1.55)	***	**	***
	Women	1.59(1.4)	2.53(1.61)	2.35(1.55)	***	*	***
	Men	1.56(1.31)	2.18(1.58)	1.99(1.53)	***	ns	***

^1^ McMenar Test; ^2^ Paired t-Student. SD: Standard deviation. *** *p* < 0.001; ** *p* < 0.01; * *p* < 0.05; ns: non-significant (*p* > 0.05) *n* = 436.

**Table 3 ijerph-20-02340-t003:** Waking headaches and bruxism.

Parameters (Yes %)		Pre-Lockdown	Lockdown	Post-Lockdown	*p*-Value		
					Pre vs. Lockdown	Lockdown vs. Post	Pre vs. Post Lockdown
Wake up with headache ^1^	Total	17.2	31.7	29.4	***	ns	***
	Women	20.3	36.7	33.1	***		***
	Men	9.9	19.8	20.6	**		**
Sleep teeth grinding/clenching ^1^	Total	31.4	35.6	36.7	**	ns	***
	Women	36.7	41.3	43.6	**		***
	Men	19.1	22.1	20.6	ns		ns
Awake teeth grinding/clenching ^1^	Total	29.8	30.7	36.0	ns	**	***
	Women	30.2	32.1	39.0		**	***
	Men	29.0	27.5	29.0		ns	ns

^1^ McMenar Test. *** *p* < 0.001; ** *p* < 0.01; ns: non-significant (*p* > 0.05). *n* = 436.

**Table 4 ijerph-20-02340-t004:** Headaches.

Diagnosis (%)		Pre-Lockdown	Lockdown	Post-Lockdown	*p*-Value		
					Pre vs. Lockdown	Lockdown vs. Post	Pre vs. post Lockdown
Infrequent Episodic TTH ^1^	Total	12.8	12.8	13.8	ns	ns	ns
	Women	12.5	13.4	14.8			
	Men	13.7	11.5	11.5			
Frequent Episodic TTH ^1^	Total	8.9	14.7	10.8	***	*	ns
	Women	11.1	15.7	11.5	*	*	ns
	Men	3.8	12.2	9.2	**	ns	*
Migraine ^1^	Total	13.5	18.1	17.0	**	ns	*
	Women	16.1	21.3	21.6	*		*
	Men	7.6	10.7	6.1	ns		ns
Other headaches (nondiagnosable) ^1^	Total	20.9	29.6	28.0	***	ns	***
	Women	24.6	32.8	29.5	**		*
	Men	12.2	22.1	24.4	**		**
None ^1^	Total	50.2	34.9	39.8	***	ns	***
	Women	43.3	27.9	32.8	***		***
	Men	64.4	51.1	52.7	***		***

^1^ McMenar Test. TTH: tension-type headache. *** *p* < 0.001; ** *p* < 0.01; * *p* < 0.05; ns: non-significant (*p* > 0.05). *n* = 436.

**Table 5 ijerph-20-02340-t005:** Temporomandibular disorders (DC-TMD).

Parameter		Pre-Lockdown	Lockdown	Post-Lockdwon	*p*-Value		
					Pre vs. Lockdown	Lockdown vs. Post	Pre vs. Post Lockdown
Joint pain ^1^ (%)	Total	30.5	34.2	32.3	ns	ns	ns
	Women	36.4	41.3	38.0			
	Men	16.8	17.6	19.1			
Joint pain intensity ^2^ (score 0–10) (mean[SD])	Total	3.61(2.24)	4.3(2.59)	3.74(2.38)	**	*	ns
	Women	3.72(2.22)	4.5(2.56)	3.96(2.37)	**	*	
	Men	3.04(2.27)	3.39(2.58)	2.84(2.26)	ns	ns	
Noises ^1^ (%)	Total	32.3	37.8	33.7	*	*	ns
	Women	38.4	43.0	38.0	*	*	
	Men	18.3	26.0	23.7	**	ns	
Blocking ^1^ (%)	Total	15.1	12.4	13.5	ns	ns	ns
	Women	17.7	14.8	16.4			
	Men	9.2	6.9	6.9			
Muscle pain ^1^ (%)	Total	28.7	33.5	30.3	*	ns	ns
	Women	33.4	39.0	35.1	*		
	Men	17.6	20.6	19.1	ns		
Muscle pain intensity ^2^ (mean[SD])	Total	3.99(2.34)	4.30(2.71)	4.13(2.49)	ns	ns	ns
	Women	4.19(2.36)	4.48(2.72)	4.47(2.51)			
	Men	3.08(2.04)	3.50(2.52)	2.81(1.97)			

^1^ McMenar test; ^2^ t-Student test. SD: Standard deviation. *n* = 436. ** *p* < 0.01; * *p* < 0.05; ns: non-significant (*p* > 0.05)

**Table 6 ijerph-20-02340-t006:** Associations between pairs.

	TMD-J	TMD-M	Headache	Bears	Well-Being
TMD-J	-----	0.77	0.17	0.29	−0.27
TMD-M	(0.71, 0.84)	-----	0.23	0.33	−0.31
Headaches	(0.09, 0.27)	(0.14, 0.32)	-----	0.43	−0.29
Bears	(0.21, 0.38)	(0.24, 0.42)	(0.36, 0.51)	-----	−0.50
Well-being	(−0.36, −0.18)	(−0.39, −0.23)	(−0.38, −0.20)	(−0.57, −0.43)	-----

The Spearman rho between variables is shown above the diagonal. 95% confidence intervals are shown below the diagonal. TMD-J: joint pain; TMD-M: muscle pain.

## Data Availability

The datasets used and/or analyzed during the current study are available from the corresponding author on reasonable request.

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
