# Peer review of "How Confinement and Back to Normal Affected the Well-Being and Thus Sleep, Headaches and Temporomandibular Disorders"

_ijerph, 2023, doi:10.3390/ijerph20032340_

Round 1

Reviewer 1 Report

Dear Authors, 

thank you for giving me the opportunity to revise the manuscript entitled "How confinement and back to normal affected the well-being and thus sleep, headaches and temporomandibular disorders." The paper aim  to assess confinement and new normal impact on well-being, sleep, headaches and temporomandibular disorders during COVID-19 pandemia. The topic is very interesting in field, giving an epidemiological aspect of TMD during a poor health care period for other disease.  The manuscript is succinct and well written. Neverthless, there are some critical issue to address:

Introduction: As mentioned by the authors,  the TMD is a complex multifactorial condition that can show different clinical condition such as Neck pain, arthralgia, and nerve injury, can occur also after surgical surgical treatment. I suggest to authors to stress this point, helping with the lecture of intro part of "Ferrillo M, Curci C, Roccuzzo A, Migliario M, Invernizzi M, de Sire A. Reliability of cervical maturation compared to hand-wirst for skeletal maturation assessment in growing subjects: A systematic review", J Back Musculoskeletal Rehabilitation, 2021" and "Molinero-Mourell P,  Ferrillo M, Cobo-Vazquez C, Sanchez-Labrador L., Ammendolia A, Migliario M, de Sire A. Type I Collagen-Based Devices to treat Nerve injuries after Oral Surgery Procedures. A systematic Review. Applied Sciences, 2021.  Moreover, the biopsychosocial model of the TMD is a very interesting point, but it would be desirable add information about the role of International classification of functioning (ICF) and possible rehabilitative approach to TMD. Please, read Ferrillo M, Ammendolia A, Paduano S, Calafiore D, Marotta N, Migliario M, Fortunato L, Giudice A, Michelotti A, de Sire A. Efficacy of rehabilitation on reducing pain in muscle-related temporomandibular disorders: A systematic review and meta-analysis of randomized controlled trials. J Back Musculoskelet Rehabil. 2022 

Methods: How have you calculated the sample size? please, specify this point. Moreover, it would be desirable to attach the full survey in appendix. 

Results: The authors affirm that 436 people have answered. How have you reached this number? please, specify this point. Moreover, a demographic table should be add

Discussion: the discussion is well written and is coherent with the results. The limits and strength of the studies are adequately described. Nevertheless, I suggest to give a possible solution for future approach in well-being monitoring (telemedicine e.g.). 

Best Regards

Author Response

Response to Reviewer 1 Comments.

Thank you very much for your interesting comments. Your comments are helping us to improve and to enrich the final version of the paper.

Point 1:

Introduction: As mentioned by the authors,  the TMD is a complex multifactorial condition that can show different clinical condition such as Neck pain, arthralgia, and nerve injury, can occur also after surgical surgical treatment. I suggest to authors to stress this point, helping with the lecture of intro part of "Ferrillo M, Curci C, Roccuzzo A, Migliario M, Invernizzi M, de Sire A. Reliability of cervical maturation compared to hand-wirst for skeletal maturation assessment in growing subjects: A systematic review", J Back Musculoskeletal Rehabilitation, 2021" and "Molinero-Mourell P,  Ferrillo M, Cobo-Vazquez C, Sanchez-Labrador L., Ammendolia A, Migliario M, de Sire A. Type I Collagen-Based Devices to treat Nerve injuries after Oral Surgery Procedures. A systematic Review. Applied Sciences, 2021.

Response 1: We have stressed this point. We add more information about the multifactorial condition of TMD. We change this paragraph. We add one of your proposed references (Ferrillo M, Curci C, Roccuzzo A, Migliario M, Invernizzi M, de Sire A. Reliability of cervical vertebral maturation compared to hand-wrist for skeletal maturation assess-ment in growing subjects: A systematic review. J Back Musculoskelet Rehabil. 2021 Apr 27. doi: 10.3233/BMR-210003).

The new paragraph is:

“TMD are a group of painful and/or dysfunctional conditions of the masticatory muscles, temporomandibular joint (TMJ) and associated structures [17,18]. TMD have multifactorial etiology, including as potential risk factors: a prolonged use of mastication muscles, grinding and clenching, malocclusion, repetitive trauma at the TMJ, psychological disorders, cervical posture or even the position of the cervical spine [17-19].”

Point 2:

Moreover, the biopsychosocial model of the TMD is a very interesting point, but it would be desirable add information about the role of International classification of functioning (ICF) and possible rehabilitative approach to TMD. Please, read Ferrillo M, Ammendolia A, Paduano S, Calafiore D, Marotta N, Migliario M, Fortunato L, Giudice A, Michelotti A, de Sire A. Efficacy of rehabilitation on reducing pain in muscle-related temporomandibular disorders: A systematic review and meta-analysis of randomized controlled trials. J Back Musculoskelet Rehabil. 2022

Response 2:

We add more information about the biopsychosocial model. And we add your proposed reference in the discussion section.

Point 3:

Methods: How have you calculated the sample size? Please, specify this point. Moreover, it would be desirable to attach the full survey in appendix.

Response 3:

To calculate the sample size, we used the following formula:

n= sample size

Zα= confidence level

N= Population size

p= probability

q= 1-p

e= error (5%)

The reason we have not included the entire survey in an appendix is because we have used standard validated surveys (Well-being index; the BEARS sleep screening tool; the International Headache Society (IHS) diagnostic criteria for TTH and migraine; DC / TMD tool and the JFLS-8 test). All these testes are referred in the methods section.

Following your recommendations we have included the bruxism questions in the material and method because there is no standard test for this purpose.

Point 4:

Results: The authors affirm that 436 people have answered. How have you reached this number? please, specify this point. Moreover, a demographic table should be add.

Response 4:

The survey was done online, when we exceeded the minimum calculated sample size (383), we discontinued the questionnaire, which remained at a total of 436.

Demographic data are explained in the text, we believe a table will repeat data. But if you think is better a table, we can eliminate the text and add a table.

Point 5:

I suggest to give a possible solution for future approach in well-being monitoring (telemedicine e.g.).

Response 5:

We add this comment at the end of the discussion.

Reviewer 2 Report

The results regarding the temporomandibular joint (TMJ) were based on a questionnaire and non a clinical examinations, which is not written in the limitations and conclusions. 

Author Response

Point 1: The results regarding the temporomandibular joint (TMJ) were based on a questionnaire and non a clinical examinations, which is not written in the limitations and conclusions.

Response 1: The limitation has been included at the end of the discussion.

“Other limitation of the study was that the survey was done online and a clinical examination was not carried out to detect and corroborate the results obtained”.

Reviewer 3 Report

Dear Authors,

The objective of this study was to assess the effects of the lockdown on wellbeing, sleep, headaches and temporomandibular disorders among a Spanish university population.

The study was in line with the aims of the journal. 

However, there are some issues that should be addressed.

Abstract

Line 19. Please, do not start the sentence with a number.

Introduction

·      Lines 1-44. Please add references.

·      I suggest improving the introduction section on TMD, headache, and sleep disorders.

·      Please refer to the Diagnostic Criteria for TMD (DC/TMD) Axis I to classify TMD. Thus, report that TMD could be divided in Group I: muscle disorders (including myofascial pain with and without mouth opening limitation); Group II: including disc displacement with or without reduction.

·      Report temporomandibular disorders (TMD) also in the Introduction.

·      Moreover, improve epidemiological data on TMD and headache, and report the correlation between them (please refer and discuss to Prevalence of temporomandibular joint disorders: a systematic review and meta-analysis. Clin Oral Investig. 2021 Feb;25(2):441-453. doi: 10.1007/s00784-020-03710-w. and Temporomandibular disorders and neck pain in primary headache patients: a retrospective machine learning study. Acta Odontol Scand. 2022 Jul 29:1-7. doi: 10.1080/00016357.2022.2105945).

·      Please, a native English should revise the paper.

·      Please, better define the gap in the scientific literature and the rationale of the study.

Materials and Methods

·      Please report that you included both myogenous (Group I) and arthrogenous TMD (Groups II and III) according to DC/TMD. 

·      migraine11?

Discussion 

Compared to arthrogenous TMD, which appear to be a localized phenomenon, myogenous TMD may present overlapping features with other disorders, such as fibromyalgia and primary headaches, characterized by chronic primary pain related to dysfunction of the central nervous system, probably through the phenomenon of central sensitization (please cite and refer to Pain Management and Rehabilitation for Central Sensitization in Temporomandibular Disorders: A Comprehensive Review. Int J Mol Sci. 2022 Oct 12;23(20):12164. doi: 10.3390/ijms232012164. PMID: 36293017; PMCID: PMC9602546 and Vale Braido GVD, Svensson P, Dos Santos Proença J, Mercante FG, Fernandes G, de Godoi Gonçalves DA. Are central sensitization symptoms and psychosocial alterations interfering in the association between painful TMD, migraine, and headache attributed to TMD? Clin Oral Investig. 2022 Nov 16. doi: 10.1007/s00784-022-04783-5. Epub ahead of print. PMID: 36383296.). Report in the discussion this point, as the different etiopathogenesis of myogenous and arthrogenous TMD should be taken into consideration; moreover, I suggest to report in the study limitation that all subtypes of TMD were included in the present paper.

Author Response

Response to Reviewer 3 Comments.

Thank you very much for your interesting comments. Your comments are helping us to improve the final version of the paper.

Point 1: Abstract. Line 19. Please, do not start the sentence with a number.

Response 1: We correct line 19.

Point 2: Lines 1-44. Please add references.

Response 2: We add new references.

Point 3: I suggest improving the introduction section on TMD, headache, and sleep disorders.

Response 3:

Point 4:  Please refer to the Diagnostic Criteria for TMD (DC/TMD) Axis I to classify TMD. Thus, report that TMD could be divided in Group I: muscle disorders (including myofascial pain with and without mouth opening limitation); Group II: including disc displacement with or without reduction.

Response 4: We used the DC/TMD and in this classification is the disorder of myofascial pain with limited opening, as described in the RDC/TMD, is not included. This is the reason why we did not include this term.

Point 5: Report temporomandibular disorders (TMD) also in the Introduction.

Response 5: The TMD was treated whit more detail in the introduction.

Point 6: Moreover, improve epidemiological data on TMD and headache, and report the correlation between them (please refer and discuss to Prevalence of temporomandibular joint disorders: a systematic review and meta-analysis. Clin Oral Investig. 2021 Feb;25(2):441-453. doi: 10.1007/s00784-020-03710-w. and Temporomandibular disorders and neck pain in primary headache patients: a retrospective machine learning study. Acta Odontol Scand. 2022 Jul 29:1-7. doi: 10.1080/00016357.2022.2105945).

Response 6: The epidemiological data on TMD and headache was improved. We included one of the recommended articles.

Point 7: Please, a native English should revise the paper.

Response 7: We will send the article for English revision when all the corrections proposed by the reviewers have been done.

Point 8: Please, better define the gap in the scientific literature and the rationale of the study.

Response 8:

Point 9: Materials and Methods. Please report that you included both myogenous (Group I) and arthrogenous TMD (Groups II and III) according to DC/TMD.

Response 9: Whe have included this information.

Point 10: migraine11?

Response 10: The reference “11” has been corrected.

Point 11: Discussion

Compared to arthrogenous TMD, which appear to be a localized phenomenon, myogenous TMD may present overlapping features with other disorders, such as fibromyalgia and primary headaches, characterized by chronic primary pain related to dysfunction of the central nervous system, probably through the phenomenon of central sensitization (please cite and refer to Pain Management and Rehabilitation for Central Sensitization in Temporomandibular Disorders: A Comprehensive Review. Int J Mol Sci. 2022 Oct 12;23(20):12164. doi: 10.3390/ijms232012164. PMID: 36293017; PMCID: PMC9602546 and Vale Braido GVD, Svensson P, Dos Santos Proença J, Mercante FG, Fernandes G, de Godoi Gonçalves DA. Are central sensitization symptoms and psychosocial alterations interfering in the association between painful TMD, migraine, and headache attributed to TMD? Clin Oral Investig. 2022 Nov 16. doi: 10.1007/s00784-022-04783-5. Epub ahead of print. PMID: 36383296.). Report in the discussion this point, as the different etiopathogenesis of myogenous and arthrogenous TMD should be taken into consideration; moreover, I suggest to report in the study limitation that all subtypes of TMD were included in the present paper.

Response 11: The proposed text and references were included.

Round 2

Reviewer 3 Report

Authors modified the text according to the suggestions. I found this work impactful and it fits well with in the scope of this journal.

In my opinion, it is suitable for publication on your Journal.